# Thymoquinone Plus Immunotherapy in Extra-Pulmonary Neuroendocrine Carcinoma: Case Series for a Novel Combination

Amr Mohamed [1,*], Asfar S. Azmi [2], Sylvia L. Asa [3], Sree Harsha Tirumani [4], Amit Mahipal [1], Sakti Cjakrabarti [1], David Bajor [1], J. Eva Selfridge [1] and Ahmed O. Kaseb [5]

1   Division of Hematology and Medical Oncology, Department of Medicine, University Hospitals, Seidman Cancer Center, Case Western Reserve University, Cleveland, OH 44106, USA
2   Division of Medical Oncology, Department of Medicine, Karmanos Cancer Institute, Wayne State University, Detroit, MI 48202, USA
3   Seidman Cancer Center, Department of Pathology, University Hospitals, Case Western Reserve University, Cleveland, OH 44106, USA
4   Seidman Cancer Center, Department of Radiology, University Hospitals, Case Western Reserve University, Cleveland, OH 44106, USA
5   Division of Gastrointestinal Medical Oncology, Department of Medicine, University of Texas MD Anderson Cancer Center, Houston, TX 77030, USA
*   Correspondence: amr.mohamed@uhhospitals.org; Tel.: +1-216-844-1257; Fax: +1-508-844-5234

**Abstract:** Background: Neuroendocrine neoplasms (NENs) are a heterogeneous group of cancers that had a significant increase in annual incidence in the last decade. They can be divided into well-differentiated neuroendocrine tumors (NETs) and poorly differentiated neuroendocrine carcinomas (NECs). Poorly differentiated NECs are aggressive forms of cancers with limited therapeutic options. The first line treatment of metastatic poorly differentiated NECs is similar to small cell lung cancer, with cytotoxic chemotherapy (etoposide plus platinum). Patients who progress have limited therapeutic options and poor overall survival, calling for other novel agents to combat this deadly disease. Therefore, in this article, we summarized the effects of a novel component, Thymoquinone (TQ, $C_{10}H_{12}O_2$), which is the main bioactive component of the black seed (*Nigella sativa*, Ranunculaceae family), plus immunotherapy in case series of patients with refractory metastatic extra-pulmonary NEC (EP-NEC) and one case of mixed neuroendocrine-non-neuroendocrine neoplasm (MiNEN). Methods: We report the effect of TQ plus dual immune checkpoint inhibitors (nivolumab plus ipilimumab) in four patients with poorly differentiated gastrointestinal Ep-NEC and MiNEN who progressed on cytotoxic chemotherapy. Results: This is the first case series to report the clinical activity of TQ plus dual immune checkpoint inhibitors (nivolumab plus ipilimumab) in patients with refractory metastatic EP-NEC. The four patients showed benefits with the combined regimen TQ plus dual ICPIs with durable response and exceeded the two years of progression-free survival. None of the four patients experienced significant toxicity, and all of them showed improvement in quality of life. Conclusion: The reported clinical courses suggest that combined TQ plus ICPIs is a potential promising regimen for refractory EP-NEC and MiNEN that deserves further prospective investigation.

**Keywords:** GEP-NET; thymoquinone; immunotherapy; combination therapy

## 1. Introduction

Neuroendocrine neoplasms (NENs) are a diverse group of cancers that arise from endocrine cells throughout the body, with a significant proportion from gastroenteropancreatic (GEP) origin [1]. A recent SEER (Surveillance, Epidemiology, and End Results) study showed a significant increase in the annual incidence of GEP-NENs [2]. Based on histologic differentiation and grade, NENs can be divided into well-differentiated neuroendocrine

tumors (NETs) and poorly differentiated carcinomas (NECs) [3,4]. Unlike low-grade well-differentiated NETs that are generally indolent, poorly differentiated NECs are highly aggressive malignancies. They are currently treated similarly to small cell lung cancer, with etoposide plus platinum, and most patients who progress have limited therapeutic options. Thus, there is a need for other novel agents to combat this deadly disease [5,6]. Previous data have shown that NENs are highly vascular neoplasms and angiogenesis is critically important for their growth and proliferation [7,8] They frequently express high levels of vascular endothelial growth factor (VEGF) ligand and its receptors, which are associated with tumor progression [9]. Several mechanisms include hypoxia and the activation of PI3K-AKT and Raf-MEK-ERK, which are the major pathways that mediate the secretion of VEGF and promote angiogenesis in NENs [10]. Genomic studies have identified a frequent loss of DAXX/ATRX in high-grade, well-differentiated NETs and TP53 mutations and Rb1 loss in poorly differentiated NECs [11]. However, these aberrations are thus far not targetable. Unfortunately, in the second-line setting, after progression on platinum/etoposide, there are no standard of care options for patients with high-grade poorly-differentiated NECs, resulting in poor survival outcomes of less than 17 months. Currently, immune checkpoint inhibitors (ICPIs) for patients with NECs are limited. Recent data from the DARTSWOG1609 and CA-209 trials suggested a potential benefit of the combination of nivolumab and ipilimumab in high-grade NENs [12,13]. However, the DART trial did not distinguish between high-grade well-differentiated NETs and poorly differentiated NECs, which are two distinct diseases, and the CA-209 trial included only two patients with poorly differentiated NECs. Developing effective therapies for refractory poorly differentiated NECs represents a serious unmet need in the clinic, and novel therapeutics are urgently needed for this highly recalcitrant disease.

Thymoquinone (TQ, C10H12O2) is the main bioactive component of the black seed (*Nigella sativa*, Ranunculaceae family), and has anti-oxidant, anti-angiogenic effects [14]. Previous studies reported its apoptotic and anti-proliferative effects in multiple cancer types, including colon, breast, and ovarian adenocarcinoma [15,16] (Table 1). In this article, we present case series of patients with EP-NECs and MiNEN who were interested in complementary medicine and were treated with this novel regimen (TQ plus dual ICPIs) at University Hospitals Seidman Cancer Center.

**Table 1.** Summary of TQ experience in several malignancies.

| Study | Cancer Type | Model (Cell Lines/Mice) | Mechanism of Action of TQ |
|---|---|---|---|
| Kundu, J. [17] | Colorectal Cancer | HCT116 Cell lines | ▪ Induces apoptosis by upregulating Bax and inhibiting Bcl-2, as well as the activation of caspases -9, -7, and -3 <br> ▪ Inactivation of STAT3 by blocking the JAK2- and Src-mediated phosphorylation of EGF receptor tyrosine kinase |
| Gali-Muhtasib, H. [16] | Colorectal Cancer | HCT-116 Cell lines | ▪ Induces apoptosis with a marked increase in p53 and p21WAF1 protein levels <br> ▪ Significant inhibition of anti-apoptotic Bcl-2 protein. |
| El-Baba [18] | Colorectal Cancer | HCT116w, DLD-1, HT29 Cell lines | ▪ Control cancer growth and interferes with RAF/MEK/ERK1/2 pathway |

**Table 1.** *Cont.*

| Study | Cancer Type | Model (Cell Lines/Mice) | Mechanism of Action of TQ |
|---|---|---|---|
| Torres, M.P. [19] | Pancreatic | FG/COLO357, CD18/HPAF Cell lines | ■ Induces apoptosis by the activation of JNK and p38 MAPK pathways<br>■ Downregulates MUC4 expression |
| Relles, D. [20] | Pancreatic | AsPC-1 and MiaPaCa-2 Cell lines and Xenografts | ■ Induces apoptosis through the upregulation of p53 and the downregulation of Bcl-2<br>■ Induces H4 acetylation and reduced HDACs expression |
| Xu, D. [21] | Cholangiocarcinoma | TFK-1, HuCCT1 Cell lines | ■ Downregulates PI3K/Akt |
| Zhu, W.Q. [22] | Gastric Cancer | HGC27, BGC823, SGC7901 Cell lines | ■ Inhibits STAT3 phosphorylation, associated with reduction in JAK2 and c-Src activity<br>■ Inhibit Bcl-2, cyclin D, survivin, and VEGF |
| Ashour, A.E. [23] | Hepatocellular Carcinoma (HCC) | HepG2 Cell lines | ■ Stimulates the expression of pro-apoptotic Bcl-xS and TRAIL death receptors<br>■ Inhibits the expression of the anti-apoptotic gene Bcl-2, as well as inhibits NF-κB and IL-8 |
| Shah, J. [24] | Hepatocellular Carcinoma (HCC) | HepG2 and SMMC-7721 cell lines | ■ Induce apoptosis through the upregulation of p53 |
| Lang, M. [25] | Small Intestine | APC$^{Min}$ mice | ■ Interferes with polyp progression in Apc$^{Min}$ mice through the induction of tumor cell-specific apoptosis<br>■ Modulating Wnt signaling through the activation of GSK-3β. |

TQ: Thymoquinone. Min: Multiple intestinal neoplasia.

## 2. Case Series

These are three patients with metastatic poorly differentiated EP-NEC and one patient with MiNEN who were concomitantly taking over-the-counter thymoquinone-based black seed oil (BSO) capsules (three capsules of 500 mg orally per day) at the time that they were being treated with dual ICPIs (nivolumab plus ipilimumab) in the second-line setting. All patients were refractory to cytotoxic chemotherapy. The data were collected in accordance with an IRB-approved retrospective data analysis protocol at UH Seidman Cancer Center.

### 2.1. Case 1

A 77-year-old male with a past medical history of anxiety and coronary artery disease, post-coronary artery bypass graft surgery (CABG) in 1991, was diagnosed with a poorly differentiated NEC of an unknown primary site. The diagnosis was formulated on biopsies of an axillary lymph node and liver mass. The patient was found to have left axillary lymphadenopathy on a screening lung CT scan. Biopsy showed a poorly

differentiated NEC (Ki-67 > 90%). Immunohistochemical analysis showed that the tumor cells expressed chromogranin, CAM5.2, and CK20 but were negative for TdT and PAX5; the Ki67 was >90%. Next-generation sequencing (NGS) of the tumor confirmed TP53 and Tet 2 mutation, the loss of Rb1, Microsatellite stable (MSS), and high tumor mutational burden (TMB: 53.2 m/MB). Skin mapping was performed and there was no evidence of a Merkel cell located. Anatomical scans with a CT and MRI of the abdomen and pelvis were performed to complete staging and reported extensive liver metastases, gastrohepatic, peri-portal, para-aortic lymphadenopathy, diffuse spinal (thoracic, lumbar), and pelvic osseous metastases consistent with metastatic NEC. Therapy was initiated with systemic cisplatin and Etoposide for two cycles complicated by fatigue and myelosuppression. Restaging scans after a second cycle showed disease progression, so second-line therapy was started with dual ICPIs in the form of nivolumab 240 mg intravenous (i.v) every 2 weeks and ipilimumab 1 mg/kg i.v. every 6 weeks. The patient was interested in holistic medicine and started taking black seed oil capsules, 500 mg, three tablets daily (1500 mg) at the same time as the second-line therapy was started. Restaging after two cycles showed a significant response in liver lesions, lymph nodes, and osseous lesions (70% shrinkage per RECIST 1.1) (Figure 1). The patient also had significant clinical improvement including weight gain of more than 10 pounds over a four-month period and improved energy levels. The treatment was well tolerated with no significant side effects. The patient finished four cycles of nivolumab plus ipilimumab and thymoquinone BSO tablets, and a restaging scan after cycle 4 showed continuing partial response. This patient remains on maintenance treatment with nivolumab 240 mg i.v. every 2 weeks and TQ-based BSO tablets, and at 24 months, he continues to have no side effects and an excellent quality of life, with the last restaging scans continuing to show partial response.

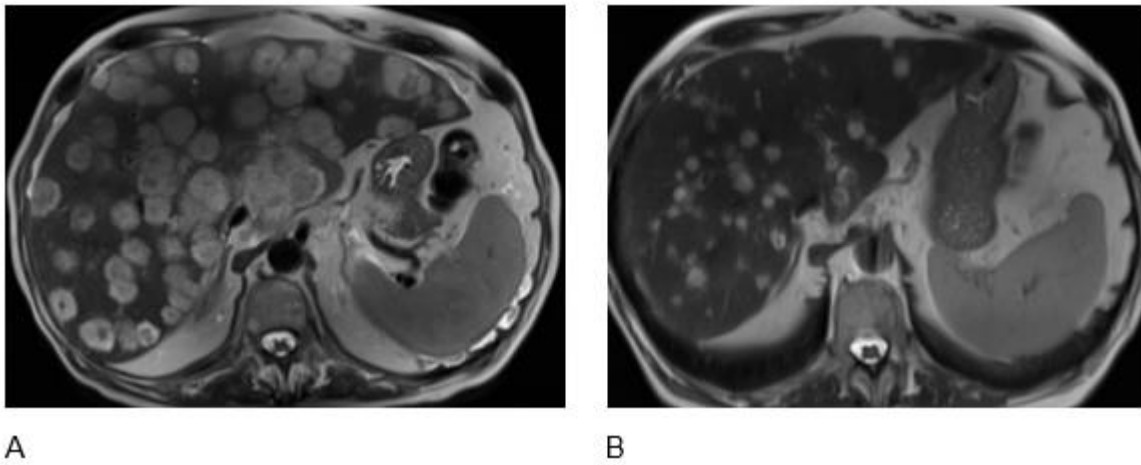

**Figure 1.** A 77-year-old man with metastatic neuroendocrine carcinoma. Axial T2-weighted MR images before (**A**) and after 4 cycles of therapy (**B**) demonstrate a significant decrease in multiple liver metastases.

*2.2. Case 2*

A 75-year-old man with a past medical history of chronic obstructive pulmonary disease (COPD), polymyalgia rheumatica (PMR), and hyperlipidemia was diagnosed with metastatic poorly differentiated NEC of the gall bladder. In June 2020, this patient presented to his primary care physician with complaints of abdominal pain, fatigue, and weight loss. An abdominal ultrasound identified an intrahepatic biliary mass; MRI confirmed the presence of a gall bladder mass that measured 8.5 × 10.3 × 6.2 cm and eroded into substantial portions of the posterior segment of the right hepatic lobe and involved a peripancreatic lymph node. A biopsy of the right liver mass was diagnosed as poorly differentiated NEC, small cell type. The malignant cells were immunohistochemically positive for cytokeratin CAM 5.2, synaptophysin, and CD56, and negative for cytokeratin AE1/3,

chromogranin, and CD20. Ki-67 was 85%. NGS confirmed mutant PTEN, TP53, Rb1 intact, MSI-H, and high TMB (53.2 m/MB). This patient was started on Cisplatin and Etoposide and finished three cycles but could not tolerate the treatment due to worsening abdominal pain, nausea, vomiting, and weight loss. Restaging scans showed disease progression in both the primary tumor invading the right hepatic lobe and the peripancreatic lymph node. Therapy was switched to dual ICPIs consisting of nivolumab 3 mg/kg i.v and ipilimumab 1 mg/kg i.v. every 3 weeks as second-line therapy. The patient started taking BSO capsules, 500 mg, three tablets daily (1500 mg). After two cycles of dual ICPI, restaging scans showed a significant response in both the primary tumor and the peripancreatic lymph node (60% shrinkage change per RECIST 1.1) (Figure 2). The treatment was well-tolerated for a total of four cycles every 3 weeks with no significant side effects. The patient reported significant improvement in his energy levels, and he gained more than 10 pounds during the first 3 months of treatment. Restaging PET-CT scans demonstrated a complete response, and he is currently on maintenance therapy with nivolumab 240 mg i.v. every 2 weeks and TQ-based BSO tablets for 22 months with no evidence of new lesions.

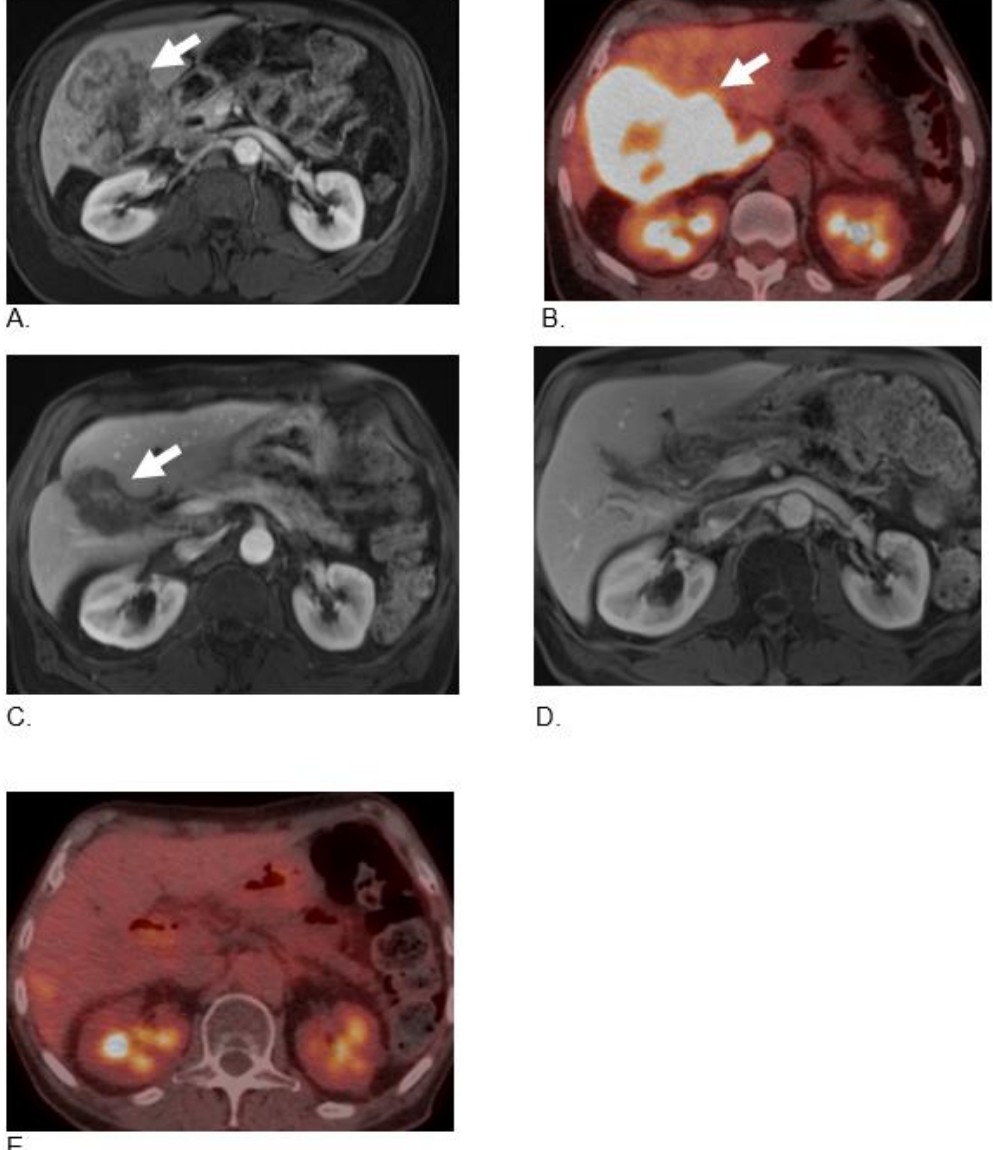

**Figure 2.** A 75-year-old male with metastatic neuroendocrine carcinoma. Axial post-contrast fat-suppressed T1-weighted (**A**) and axial-fused 18F-FDG-PET/CT (**B**) images demonstrate large heterogeneous mass in the right lobe of the liver with intense metabolic activity (arrow). Follow-up MRI

after four cycles of therapy (**C**) demonstrates a significant decrease in the liver tumor with a significant decrease in enhancement (arrow). Follow-up MRI (**D**) and 18F-FDG-PET/CT (**E**) 2 years after baseline imaging on maintenance therapy shows complete resolution of mass and metabolic activity.

*2.3. Case 3*

A 67-year-old male with a past medical history of stage III B who has been diagnosed with Colon MiNEN (mixed neuroendocrine-nonneuroendocrine carcinoma with adenocarcinoma component). A restaging anatomical scan at that time excluded metastatic disease, and the patient had left laparoscopic hemicolectomy followed by adjuvant mFOLFOX chemotherapy. On surveillance, follow-up restaging scans showed new areas of metastatic disease in the liver, peritoneum, and abdominal lymph nodes, which were all FDG avid on a PET/CT scan. Therefore, he underwent a liver biopsy, which confirmed metastatic poorly differentiated NEC. Immunohistochemical analysis showed that the tumor cells expressed chromogranin, synaptophysin, INSM-1, and CDX-2 consistent with NEC of gastrointestinal origin. Next-generation sequencing of the tumor confirmed TP53 mutation, the loss of Rb1, Microsatellite stable (MSS), and low TMB (6.3 m/MB). The patient started on carboplatin and etoposide and finished a total of six cycles. After three cycles, a restaging scan showed a partial response, but after cycle six scans, it showed disease progression in both liver and peritoneal metastases.

Therapy was switched to dual ICPIs, nivolumab 3 mg/kg i.v and ipilimumab 1 mg/kg i.v. every 3 weeks; in addition, he began to ingest TQ-based BSO (one teaspoon of liquid form which is equivalent to 1500 mg oral tablets). After completing four cycles of immunotherapy and BSO, restaging scans demonstrated significant improvement in metastatic disease with decreased conspicuity of previously seen upper abdominal lymph nodes, and peritoneal nodules, and neither of the liver lesions was any longer demonstrable (Figure 3). The patient had overall complete response and was started on maintenance TQ-based BSO and nivolumab 240 mg i.v. every 2 weeks with no evidence of progression yet (PFS of 10 months).

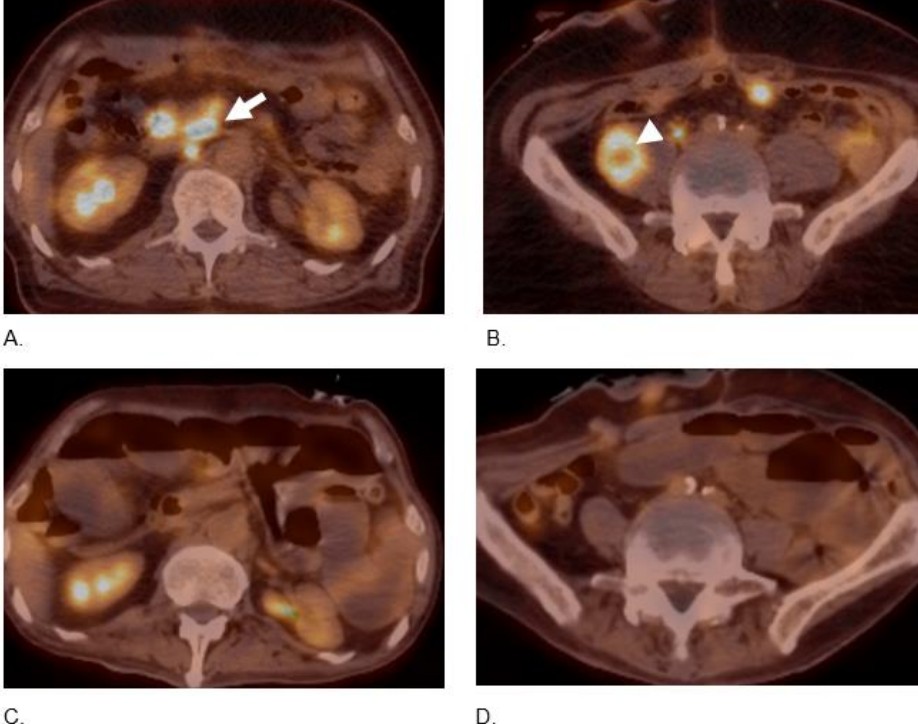

**Figure 3.** (**A**,**B**) 67-year-old male with metastatic colon MiNEN/NEC. Axial fused 18F-FDG-PET/CT image at baseline demonstrates multiple hypermetabolic retroperitoneal lymph nodes (arrow) and

hypermetabolic right psoas mass (arrowhead). (**C,D**) Follow-up 18F-FDG-PET/CT 14 months later demonstrates the complete resolution of nodes and right psoas mass.

*2.4. Case 4*

An 85-year-old male with a past medical history of hypertension and coronary artery disease, CAD, BPH, and HTN, was diagnosed with a colonic MiNEN (70% poorly differentiated NEC and 30% adenocarcinoma components). Staging CTs of the chest, abdomen, and pelvis excluded metastatic disease; therefore, he underwent laparoscopic right hemicolectomy; final pathology staging was T4N2bM0 with 50% poorly differentiated mixed small cell and large cell features (Ki67 95%) and a 50% adenocarcinoma component. NGS confirmed mutant BRAF, TP53, Rb1 loss, MSI-H, and high TMB (43.7 m/MB). The patient refused platinum-etoposide as adjuvant therapy due to side effects, and instead, he agreed to be treated for 6 months with adjuvant mFOLFOX. After a 6-month course of adjuvant therapy with mFOLFOX, a restaging scan and PET-FDG showed disease progression with mainly an enlargement of the retroperitoneal, aortocaval, gastrohepatic, and right retrocrural lymph nodes. The therapy was switched to carboplatin and etoposide, and he finished six cycles with an initial good response after both cycle three and cycle six. However, further restaging imaging showed disease progression in less than six months from the last cycle of chemotherapy. Therefore, he was switched to dual ICPIs, nivolumab 3 mg/kg i.v., and ipilimumab 1 mg/kg i.v. every 3 weeks. In addition, he began to ingest TQ-based BSO capsules, 500 mg, three tablets daily (1500 mg). After completing two cycles of immunotherapy, restaging scans demonstrated significant improvement in retroperitoneal, aortocaval, and liver metastases (Figure 4). Repeated restaging scans after completing the three cycles continue to show treatment response with a significant decrease in the size of the retroperitoneal and aortocaval lymph nodes and no evidence of liver lesions. The patient maintained with no evidence of progression for 12 months, but he refused to resume maintenance therapy due to the development of immunotherapy-related pneumonitis. Recent restaging scans showed disease progression with new portocaval lymphadenopathy but he was not on any active therapy (Table 2).

**Table 2.** Summary of TQ –dual ICPIs combined regimen in neuroendocrine carcinoma.

| Primary Tumor | Histological Differentiation | Patient Age at Diagnosis (Years) | Gender | First Line Treatment | Second Line Treatment | Maintenance Therapy | Outcome |
|---|---|---|---|---|---|---|---|
| Unknown Primary | EP Poorly differentiated neuroendocrine carcinoma (Small cell) | 77 | M | Carboplatin and Etoposide | Nivolumab 240 mg intravenous (i.v) every 2 weeks and Ipilimumab 1 mg/kg i.v. every 6 weeks for 4 cycles plus TQ-BSO (3 tablets 500 mg daily) | Nivolumab 240 mg i.v. every 2 weeks, plus BSO (TQ) tablets 1500 mg daily | Alive with PR and PFS of 24 mos |
| Gall Bladder | Poorly differentiated neuroendocrine carcinoma (Small cell) | 75 | M | Carboplatin and Etoposide | Nivolumab 3 mg/kg i.v and Ipilimumab 1 mg/kg i.v. every 3 weeks for 4 cycles plus TQ-BSO (3 tablets 500 mg daily) | Nivolumab 240 mg i.v. every 2 weeks, plus BSO (TQ) tablets 1500 mg daily | Alive with CR and PFS of 22 mos |
| Colon MiNEN | Mixed neuroendocrine-non-neuroendocrine neoplasm (MiNEN). | 67 | M | Carboplatin and Etoposide | Nivolumab 3 mg/kg i.v and Ipilimumab 1 mg/kg i.v. every 3 weeks for 4 cycles plus TQ-BSO (1 teaspoon oil formula daily) | Nivolumab 240 mg i.v. every 2 weeks plus BSO (TQ) tablets 1500 mg daily | Alive with CR and PFS of 10 mos |

**Table 2.** *Cont.*

| Primary Tumor | Histological Differentiation | Patient Age at Diagnosis (Years) | Gender | First Line Treatment | Second Line Treatment | Maintenance Therapy | Outcome |
|---|---|---|---|---|---|---|---|
| Colon MiNEN | Mixed neuroendocrine-non-neuroendocrine neoplasm (MiNEN). 70% small cell and 30% adenocarcinoma | 85 | M | Adjuvant mFOLFOX | Nivolumab 3 mg/kg i.v and Ipilimumab 1 mg/kg i.v. every 3 weeks for 3 cycles plus TQ-BSO (3 tablets 500 mg daily) | BSO (TQ) tablets 1500 mg daily | Alive with PR and PFS of 12 month |

M = male; CR= Complete response, EP: Extra pulmonary; mFOLFOX: Modified FOLFOX.

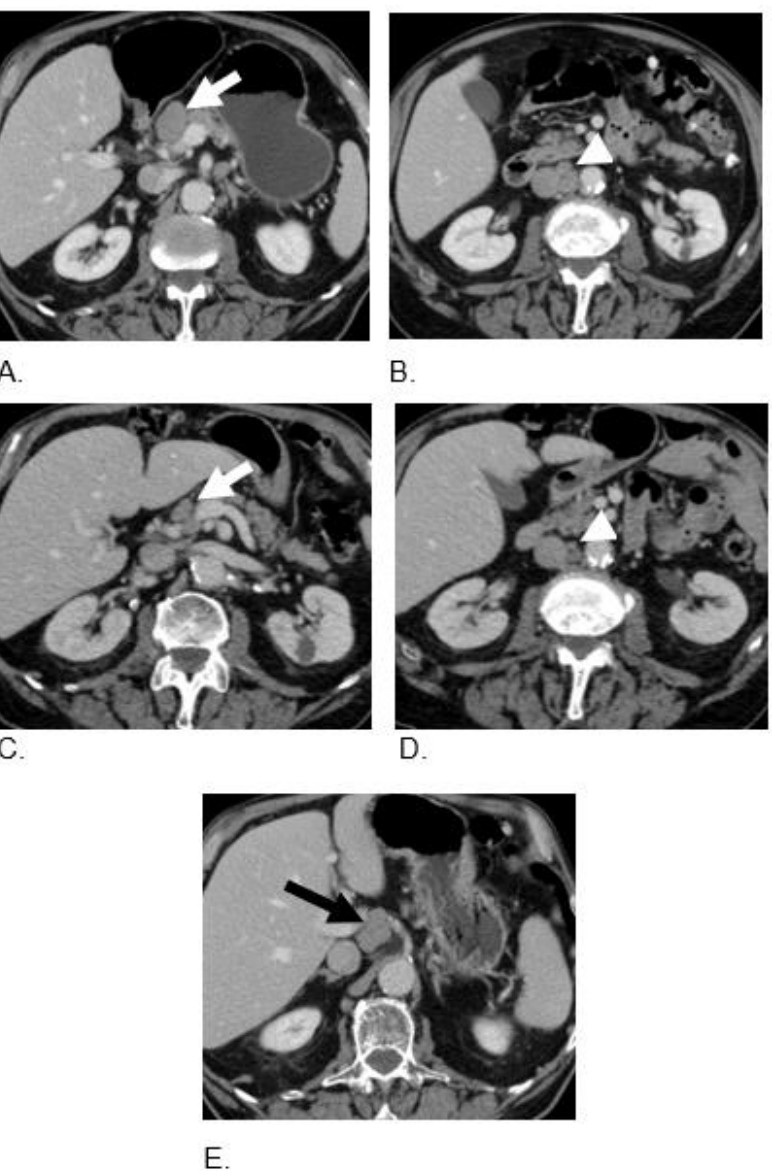

**Figure 4.** An 88-year-old male with metastatic colon MiNEN. Axial contrast enhanced CT images at baseline (**A**,**B**) demonstrate enlarged peripancreatic (arrow) and aortocaval (arrowhead) lymph nodes. Follow-up CT after three cycles of therapy (**C**,**D**) demonstrate a significant decrease in the size of the nodes (arrow, arrowhead). CT scan 12 months after therapy (**E**) shows disease progression with new portocaval lymphadenopathy (black arrow).

## 3. Discussion

EP-NECs are aggressive tumors with poor overall survival and are treated with chemotherapy regimens that were adopted from small cell lung cancer [26,27]. To date, the role of the immune checkpoint blockade with anti–CTLA-4 and anti–PD-1 in the second setting is debatable given the lack of prospective trials exclusively for poorly differentiated EP-NECs. However, recent data from the DARTSWOG1609 and CA-209 trials suggested a potential benefit of nivolumab and ipilimumab in high-grade NENs; the response rates were widely variable (RR 9–44%) and these studies did not distinguish between high-grade well-differentiated NETs and poorly differentiated NECs, so these results cannot be adopted as the standard of care [12,13]. Additionally, there were recently real-world evidence and single institution experiences which showed that a nivolumab plus ipilimumab regimen has modest activity in heavily pre-treated NECs that have progressed on prior cytotoxic chemotherapy. The results were suboptimal (RR 10–20%) compared to DART and CA209 trials, given that they included mostly poorly differentiated NECs (80%) [28,29]. Therefore, improving the efficacy of ICPIs in poorly differentiated EP-NEC is an area of unmet medical need and requires further prospective investigation.

In this article, we summarized our single institution experience at UH Seidman Cancer Center. We reviewed case series of three patients with metastatic EP-NECs and one patient with colon MiNEN who benefited from combining dual ICPIs with TQ (BSO capsules). All three had significant responses and improved performance status without any adverse side effects. This single institution experience shows encouraging results, and we were able to provide some degree of disease stabilization for patients who progressed on first-line cytotoxic chemotherapy and have limited therapeutic options.

TQ derived from Nigella sativa (commonly known as Black seed) has been extensively studied for its biological activities and pre-clinical data and points to a promising potential therapeutic option in the management of several cancers [15,16]. Previous studies reported its antitumor and anti-angiogenic effects on different cancers in vitro and in vivo [30,31] (Table 1). These studies identified a possible mechanism of the antitumor and anti-angiogenic activity of TQ by suppressing the activation of VEGF-induced ERK and AKT and their regulated downstream molecules [30]. AKT, also named protein kinase B (PKB), is a critical regulator generally involved in cell cycle and proliferation, and extracellular signal-regulated kinase (ERK) is an important factor in mediating cell proliferation and survival [32,33]. Both AKT and ERK activation are essential for angiogenesis [34,35]. Therefore, the TQ-mediated inhibition of AKT and ERK activation in endothelial cells could significantly impact angiogenesis pathways and could increase the effectiveness of immunotherapy through its antiangiogenic and immunomodulation mechanisms.

Preclinical data have shown the synergistic effect of combining antiangiogenic therapies with ICPIs. The results demonstrated that combining anti-VGEF with ICPIs reprograms the tumor immune microenvironment, increases the effectiveness of immunotherapy, and alleviates resistance [36]. In vitro, anti-angiogenesis effectively increased T-cell infiltration and supported the infiltration of immunosuppressive cells in the tumor microenvironment, enhancing the efficacy of the ICPIs [37]. In a phase IB/II trial (KEYNOTE-146), lenvatinib combined with pembrolizumab has demonstrated antitumor activity in patients with advanced endometrial cancer. This single-arm, multicenter trial showed an objective response rate (ORR) of 38.3%; importantly, most responders did not have high levels of microsatellite instability (MSI-H) [38]. Another study of the same regimen in patients with head and neck cancers showed an ORR of 36.4% regardless of PD-L1 status [39]. Recently, preliminary results from a phase II basket study showed that the combination of atezolizumab and bevacizumab demonstrated moderate clinical activity in patients with advanced NETs (RR 20%, median PFS 19.6 months) [40]. The results of this study and previous trials indicate improvement in patient outcomes and enhanced cancer immunotherapy when combined with antiangiogenic agents.

In addition to its pro-apoptotic and angiogenic effects, previous in vitro and in vivo findings reported the ability of TQ to modulate inflammation and cellular and humoral

adaptive immune responses [41]. TQ has been shown to augment the cytotoxic activity of natural killer (NK) cells against cancer cells [42,43]. Furthermore, preclinical studies provide evidence that TQ may enhance the capability of CD8+ T-cells in which the addition of low concentrations of TQ during T-cell activation resulted in the enhanced survival of the activated T-cells and sustained the expression of CD62L [44]. These results suggest that TQ has a beneficial effect in enhancing T-cell infiltration and survival, which may directly impact the tumor microenvironment and potentially improve the efficacy of ICPIs.

These previous data suggest that TQ exhibits anticancer activity via numerous mechanisms of action. Some of the proposed mechanisms of action are anti-angiogenesis and immunomodulation [30] (Figure 1). Given that NENs are highly vascular diseases, combining anti-angiogenesis with immune checkpoint inhibitors (ICPIs) may improve immunotherapy efficacy in these tumors.

The benefit of ICPIs is limited in patients with NENs, and previous efforts were made to identify predictive biomarkers. In particular, patients with MSI-H and/or high levels of TMB (defined by a median of 10 mutations per megabase (mut/Mb)), seem to benefit the most from ICPIs. In our case series, patients had complete responses whether they were MSS or MSI-H and independent of their TMB status. Therefore, our pilot study will include a biomarker analysis to identify reliable predictive factors for ICPIs in EP-NECs.

The dosing of 3 g (six tablets of 500 mg per day) was evaluated in the previous literature, including early-phase trials across a number of therapeutic indications. These trials used a higher dose of 3 g per day, reporting side effects ranging from the absence of any side effects to mild side effects; otherwise, the drug has been well tolerated. Mahdavi et al. conducted a double-blind placebo-controlled randomized clinical trial to determine the effects of BSO oil combined with a calorie-restricted diet on systemic inflammatory biomarkers in 90 obese women [45]. The intervention group received 3 g/day of BSO oil soft gel capsules and subjects reported no side effects during the intervention except mild gastrointestinal problems. In another study by Bamosa et al., subjects with type 2 diabetes mellitus were randomly divided into three groups to receive BSO extracts at 1 g, 2 g, and 3 g per day, respectively, for 12 weeks [46]. The three doses of BSO were well tolerated, with only three subjects experiencing mild epigastric discomfort that settled down after taking the capsules after meals. Recently, a randomized phase II trial in COVID patients compared adverse events for patients who took TQ (3 g per day) or a placebo, and the results indicated that TQ was safe and tolerable [47]. Our four patients took only three tablets of 500 mg per day with no documented side effects, which is correlated with these previous data.

## 4. Conclusions

This is the first report to demonstrate the clinical efficacy of combined TQ (BSO capsules) with dual ICPIs (CTLA-4/PD-1 blockade) in patients with metastatic EP-NECs who were refractory to first-line chemotherapy. The role of combining TQ with immunotherapy should be further investigated in patients with EP-NECs and MiNEN independent of the primary tumor site. Future studies should also focus on investigating correlative biomarkers for this novel regimen, including lymphocyte infiltration, MMR/TMB status, and angiogenesis biomarkers. Given our positive experience and encouraging results, we are conducting an ongoing pilot study using TQ Formula with dual ICPIs in patients with refractory GEP-NECs (NCT05262556). This is a patented enteric-coated BSO formulation with a specific tight range of TQ of about 1.6% that has been fully characterized and manufactured under good manufacturing practice (GMP) conditions to meet the US Food and Drug Administration (FDA) guidelines to conduct an investigational new drug (IND) trial in patients with refractory GEP-NECs.

**Author Contributions:** A.M. (Amr Mohamed): Designing the research work, writing the manuscript; S.L.A.: Editing the manuscript; A.O.K.: Editing the manuscript and reviewing the data; S.H.T.: Reviewing the anatomical and functional images and editing the manuscript; A.S.A.: Writing and

editing the manuscript; A.M. (Amit Mahipal), S.C., D.B., and J.E.S.: Editing the manuscript. All authors have read and agreed to the published version of the manuscript.

**Funding:** This research received no external funding.

**Institutional Review Board Statement:** Not applicable. This was not a prospective trial; this was a retrospective case report analysis of three patients.

**Informed Consent Statement:** The consent from the three patients to publish their information has been obtained.

**Data Availability Statement:** Data and materials are available upon request.

**Conflicts of Interest:** The authors declare no conflict of interest.

## Abbreviations

| | |
|---|---|
| TQ | Thymoquinone |
| NET | Neuroendocrine tumors |
| NEC | Neuroendocrine carcinoma |
| GEP | Gastroenteropancreatic |
| VEGF | Vascular endothelial growth factor |
| ERK | Extracellular-regulated kinase |

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
