# Peer review of "Thymoquinone Plus Immunotherapy in Extra-Pulmonary Neuroendocrine Carcinoma: Case Series for a Novel Combination"

_curroncol, doi:10.3390/curroncol29110707_

Round 1
Reviewer 1 Report
In this article, the authors present case series of patients with EP-NECs and MiNEN. who were interested in complementary medicine and were treated with this novel regimen (TQ plus dual ICPIs) at University Hospitals Seidman Cancer Center. I think it can be accepted with minor revision:In figure 1,-4, is there any scale bar for it?
Author Response
In figure 1,-4, is there any scale bar for it?
Thank you for the suggestion. We added the requested scale bar to the figures
Reviewer 2 Report
The case report submitted by Mohamed et al summarized the effects of a novel component Thymoquinone (TQ, C10H12O2), plus immunotherapy in case series of patients with refractory metastatic extra-pul-monary NEC (EP-NEC) and one case of mixed neuroendocrine-non-neuroendocrine neoplasm (MiNEN). The case reported presented a promising aspect for Neuroendocrine neoplasms. However, the conclusion presented in the case report need further necessary controls and experiments.
Authors have used Thymoquinone (TQ, C10H12O2) along with the dual ICPIs in one or all case studies and did not presented any results or similar previous reports where TQ solely, exerts these positive results which causes doubts on the results obtained are due to the effects of TQ or ICPIs ?
Did authors try ICPIs individually (in absence of TQ) in any of case studies and noticed similar or different changes ?
In the discussion , authors have mentioned that the TQ has been earlier used in vitro and in vivo for different cancers, however, authors did not discuss about how they have chosen dose for the case studies.?
Author Response
However, the conclusion presented in the case report need further necessary controls and experiments.
- Thank you for your comment. This was only case series and we mentioned in the discussion part that we will investigate this regimen in a pilot study and if it showed efficacy we will plan for a prospective phase II randomized study against immunotherapy alone.
Authors have used Thymoquinone (TQ, C10H12O2) along with the dual ICPIs in one or all case studies and did not presented any results or similar previous reports where TQ solely, exerts these positive results which causes doubts on the results obtained are due to the effects of TQ or ICPIs ?
- Thank you for your comment. We did not see the same results in cases who got immunotherapy alone and because this is only case series for patients who had TQ plus nivolumab plus ipilimumab and not a retrospective study we did not add these cases in our manuscript. In-addition, the real world evidence and single institution experience from both Mayo clinic and Moffitt cancer center seems to have low ORR (<20%) using immunotherapy alone. Our future trials will prospectively investigate TQ plus ICPIs vs ICPIs to explore the additive effect of TQ.
Did authors try ICPIs individually (in absence of TQ) in any of case studies and noticed similar or different changes ?
- Thank you for your comment. We treated other patients with just the NCCN recommended nivolumab plus ipilimumab regimen in the second line setting and align with the historical data we did not see the same results (ORR<20% and PFS less than six month). We did not add these cases because this is only case series for patients who had TQ plus nivolumab and ipilimumab and not a retrospective study.
In the discussion , authors have mentioned that the TQ has been earlier used in vitro and in vivo for different cancers, however, authors did not discuss about how they have chosen dose for the case studies.?
- Thank you for your comment. However, the patients took this TQ-BSO on their own we shared the recommended dose in the literature and we added this paragraph to discuss why we chose this specific dose: The dosing of 3 g (6 tablets of 500 mg per day) was evaluated on previous literature include early phase trials across a number of therapeutic indications. These trials used that higher dose of 3g per day reporting side effects ranging from absence of any side effects to mild side effects; otherwise, the drug has been well tolerated. Mahdavi et al. conducted a double-blind placebo-controlled randomized clinical trial to determine the effects of BSO oil combined with a calorie-restricted diet on systemic inflammatory biomarkers in 90 obese women. [48] The intervention group received 3g/day BSO oil soft gel capsules and subjects reported no side effects during the intervention except mild gastrointestinal problems. Another study by Bamosa et al., subjects with type 2 diabetes mellitus were randomly divided into three groups to receive BSO extracts at 1gm, 2gms and 3gms per day respectively for 12 weeks. [49] The three doses of BSO were well tolerated with only three subjects who experienced a mild epigastric discomfort that settled down after taking the capsules post meals. Recently, a randomized phase II trial in COVID patients compared adverse events for patients who took TQ (3 g per day) or placebo and the results indicated that TQ was safe and tolerable. [50] Our four patients took only three tablets of 500 mg per day with no any documented side effects, which is correlated with these previous data.
Round 2
Reviewer 2 Report
Authors have addressed the concerns raised in the previous version of the manuscript and hence I recommend it for its publication